# Plant Growth Promotion and Biological Control against *Rhizoctonia solani* in Thai Local Rice Variety “Chor Khing” Using *Trichoderma breve* Z2-03

**DOI:** 10.3390/jof10060417

**Published:** 2024-06-11

**Authors:** Warin Intana, Nakarin Suwannarach, Jaturong Kumla, Prisana Wonglom, Anurag Sunpapao

**Affiliations:** 1School of Agricultural Technology and Food Industry, Walailak University, Nakhon Si Thammarat 80161, Thailand; iwarin@wu.ac.th; 2Research Center of Microbial Diversity and Sustainable Utilization, Chiang Mai University, Chiang Mai 50200, Thailand; suwan.462@gmail.com (N.S.); jaturong_yai@hotmail.com (J.K.); 3Faculty of Technology and Community Development, Thaksin University, Papayom 93210, Thailand; prisana.w@tsu.ac.th; 4Agricultural Innovation and Management Division (Pest Management), Faculty of Natural Resources, Prince of Songkla University, Hatyai 90110, Thailand

**Keywords:** fungicidal activity, indole-3-acetic acid, local rice, plant growth promoting fungus

## Abstract

Several strains of *Trichoderma* are applied in the field to control plant diseases due to their capacity to suppress fungal pathogens and control plant diseases. Some *Trichoderma* strains also are able to promote plant growth through the production of indole-3-acetic acid (IAA). In southern Thailand, the local rice variety “Chor Khing” is mainly cultivated in the Songkhla province; it is characterized by slow growth and is susceptible to sheath blight caused by *Rhizoctonia solani*. Therefore, this research aimed to screen *Trichoderma* species with the ability to promote plant growth in this rice variety and enact biological control against *R. solani*. A total of 21 *Trichoderma* isolates were screened for indole compound production using the Salkowski reagent. The Z2-03 isolate reacted positively to the Salkowski reagent, indicating the production of the indole compound. High-performance liquid chromatography (HPCL) confirmed that Z2-03 produced IAA at 35.58 ± 7.60 μg/mL. The cell-free culture filtrate of the potato dextrose broth (CF) of Z2-03 induced rice germination in rice seeds, yielding root and shoot lengths in cell-free CF-treated rice that were significantly higher than those of the control (distilled water and culture broth alone). Furthermore, inoculation with *Trichoderma* conidia promoted rice growth and induced a defense response against *R. solani* during the seedling stage. *Trichoderma* Z2-03 displayed an antifungal capacity against *R. solani*, achieving 74.17% inhibition (as measured through dual culture assay) and the production of siderophores on the CAS medium. The pot experiment revealed that inoculation with the *Trichoderma* sp. Z2-03 conidial suspension increased the number of tillers and the plant height in the “Chor Khing” rice variety, and suppressed the percentage of disease incidence (PDI). The *Trichoderma* isolate Z2-03 was identified, based on the morphology and molecular properties of ITS, translation elongation factor 1-alpha (*tef1-α*), and RNA polymerase 2 (*rpb2*), as *Trichoderma breve* Z2-03. Our results reveal the ability of *T. breve* Z2-03 to act as a plant growth promoter, enhancing growth and development in the “Chor Khing” rice variety, as well as a biological control agent through its competition and defense induction mechanism in this rice variety.

## 1. Introduction

*Trichoderma* is a genus of fungi in the family Hypocreaceae, commonly present in all soils [1]. The *Trichoderma* species are known as fast-growing fungi for whom the appropriate temperature range is 25–30 °C; however, some species are able to grow at 45 °C [2]. At first, colonies of *Trichoderma* are transparent on corn meal dextrose agar (CMDA), or white when cultured on potato dextrose agar (PDA) [3]. Conidiophores are highly branched and loosely or compactly tufted. Phialide develop at the tips of conidiophores, and the tips of the phialide form into conidia. Conidia appear colorless to green and are variable in shape and size. Most *Trichoderma* species produce moist conidial masses. *Trichoderma* species have been known to act as plant growth promoters [4], biological control agents [5,6], and pathogens [7].

Indole-3-acetic acid (IAA) is the most common plant hormone of the auxin class, and is involved in cell division and cell elongation [8]. This ability enables it to increase the tissues and biomass of plants. IAA is not only produced in plants, as some micro-organisms are also able to produce IAA. Several strains of *Trichoderma* species promote plant growth through the production of IAA. For instance, the IAA-producing *T. harzianum* strain promotes growth through increasing the leaf area, and can also control wilt disease in tomato plants [9]. The application of PDB-cultured *T. virens* T49 and *T. harzianum* T115 yielded the greatest weights under water stress [10]. Other strains of *T. viride* can potentially be used to reduce *Fusarium* wilt and to promote the plant growth and yield in tomato plants [11].

Many species of *Trichoderma* can be used as biological control agents through their action of reducing fungal pathogens through antibiosis [12]. They also compete for nutrients and space [13], act as mycoparasites [14], induce disease resistance in plants [15], and produce volatile/non-volatile antifungal compounds [16]. For instance, *T. koningiopsis* PSU3-2 demonstrated competitiveness against the fungus *Colletotrichum gloeosporioides,* and suppressed fungal growth by releasing volatile antifungal compounds [17]. The application of the spore suspension of *T. asperelloides* PSU-P1 induced defense responses through upregulating defense-related gene expression and elevating enzymes related to defense against gummy stem blight disease, caused by the fungus *Stagonosporopsis cucurbitacearum,* in muskmelon plants [18]. *Trichoderma asperellum* strain K1-02 exhibited direct parasitism against *Neoscytalidium dimidiatum*, which causes stem canker in dragon fruits [3].

Rice (*Oryza sativa*) is the most important crop plant, and its growth accounts for a significant portion of the Thai economy and labor force. Thailand has a strong tradition of rice production for consumption and export [19]. The cultivation of rice takes place in every part of Thailand, due to the presence of suitable weather for its growth and development. In southern Thailand, local rice cultivars such as the “Chor Khing” variety are cultivated for consumption and the conservation of germplasm, and, so, are considered geographical indicator (GI) plants. However, the cultivation of local rice encounters problems related to low growth yields and infection with sheath blight disease. The use of plant growth-promoting fungi to increase growth and induce disease resistance in local rice is an alternative way to solve this problem and support sustainable agriculture. Therefore, this research aimed to screen *Trichoderma* species with the ability to promote plant growth, induce disease resistance in plants, and inhibit fungal pathogens causing sheath blight in the local “Chor King” rice variety.

## 2. Materials and Methods

### 2.1. Sources of Plant, Trichoderma, and Pathogen

“Chor Khing” rice variety seeds were obtained from the rice germplasm collection in the Phatthalung province in southern Thailand. A total of 21 isolates of *Trichoderma* were taken from soil (rhizosphere soil) obtained from the Songkhla province in southern Thailand, using dilution pour plates via the method of a previous study [20]. *Rhizoctonia solani*, which causes rice sheath blight, was obtained from the Department of Plant Pathology, Faculty of Agriculture, Khon Kaen University, Thailand. *Trichoderma* isolates and *R. solani* were cultured on potato dextrose agar (PDA) and incubated at an ambient temperature (28 ± 2 °C) for 3 days before use in the following study.

### 2.2. Screening of a Total Indole Compounds Production

To screen indole compound-producing *Trichoderma* isolates, IAA production was assessed according to the method described by Patten and Glick [21]. A total of 21 isolates of *Trichoderma* were cultured on potato dextrose broth (PDB) supplemented with 0.2% sterilized L-tryptophan solution, incubated at an ambient temperature in a rotary shaker operated at 120 rpm for 7 days. Then, the culture was centrifuged at 10,000× *g* rpm at 4 °C for 15 min, and the supernatant was collected. In total, 1 mL of cell-free supernatant was mixed with 2 mL of the Salkowski reagent (1 mL of 0.5 M FeCl_3_ in 50 mL of 35% HClO_4_) and incubated at an ambient temperature for 1 h. The development of a pink color in the reaction tube indicates the production of indole compounds. To confirm IAA production, HPLC was used to identify IAA in further experiments.

### 2.3. HPLC Analysis of Fungal IAA

Indole-3-acetic acid testing was conducted to confirm that all samples were positive in the Salkowski reagent. A total of 400 mL of culture filtrate, as described in Section 2.2, had its pH adjusted to 2.5 through the addition of 1 N HCl, and was subjected to IAA extraction using ethyl acetate in a ratio of 1:1. Then, the solvent was evaporated using a rotary evaporator to derive crude IAA. The IAA produced from a selected fungal isolate was quantified using high-performance liquid chromatography (HPLC), according to the method of Kumla et al. [22] with some modifications. HPLC analysis was performed on a Shimadzu Prominence UFLC system, coupled with a LC-40D XS pump, using an SIL-40C XS autosampler, CTO-40C column oven, SL-40 system controller, and SPD-M40 photodiode array detector (Shimadzu, Japan). The sample was separated on a Mightysil RP-18 (250 × 4.6 mm, 5 μm) column at 35 °C. The mobile phase included a solution of 80% acetonitrile in deionized water (A) and 2.5% acetic acid in deionized water at pH 3.8 (adjusted by 10 M KOH) (B). The following gradient program was used: 0–25 min, 0–20% B; 25–31 min, increased to 50% B; 31–42 min, increased to 100% B. The flow rate was 0.5 mL/min, and the detection was performed with absorption at 280. The injection volume was 5 μL. The presence of IAA was determined by comparing both the retention time and absorption spectrum with the IAA standard. The fungal IAA was quantified using the calibration curve constructed according to the IAA standard.

### 2.4. Ability of Cell-Free Culture Filtrate to Impact Seed Germination and Plant Growth

A selected *Trichoderma* isolate was cultivated in potato dextrose broth (PDB) and incubated at an ambient temperature for 7 days. The culture broth was then filtered with 0.45 μm filter paper to derive a cell-free culture filtrate (CF). The cell-free CF was diluted with sterilized distilled water (DW) in a ratio of 1:1. To test the effect of the cell-free CF on seed germination and plant growth in rice, the root length and shoot length were measured. Rice seeds were surface-disinfected with 5% sodium hypochlorite (NaOCl) and sterile distilled water (DW) three times. A total of 20 rice seeds were placed directly on the filter paper, which was soaked with the cell-free CF in a Petri dish (9 cm), whereas the control group was soaked with autoclaved PDB alone or sterilized DW alone. The experiment was conducted over three replicates and was repeated twice. The root and shoot lengths of germinations from rice seeds were measured at 4 and 5 days post-application (dpa).

### 2.5. Effect of Spore Suspension on Growth of Rice Seedlings

A selected *Trichoderma* isolate was cultured on PDA and incubated at an ambient temperature for 7 days. Conidia were harvested with sterilized DW and adjusted to a concentration of 10^6^ conidia/mL. The rice seeds were surface-disinfected by applying 5% NaOCl followed by sterilized DW three times. These rice seeds were then soaked in sterilized DW overnight. Vermiculite was mixed with peat moss in a ratio of 9:1 in 60 × 55 mm cultivated pots. Five rice seeds were grown in pots containing vermiculites and peat moss. A total of 1 mL of conidial suspension of the selected *Trichoderma* isolate was applied to each cultivated pot every 5 days, whereas the application of sterilized DW served as the control treatment. Each treatment employed 5 seeds and was conducted over 10 replications, with the experiment being repeated twice. The cultivated pots were then incubated at an ambient temperature with natural light for 10 days. The growth parameters of rice seedlings, including the shoot and root length, fresh weights of shoots and roots, and total chlorophyll content, were measured.

The total chlorophyll contents were compared between *Trichoderma*-treated and untreated (control) seedlings. A total of 100 mg of rice leaves and stems was subjected to chlorophyll extraction in 80% acetone, then incubated at 4 °C in the dark for 10 h. Chlorophyll solutions were assessed via spectrophotometry at 663 and 645 nm. The total chlorophyll content (chlorophyll a and b) was calculated using the formula:Total chlorophyll = [(8.02 × A663) + (20.2 × A645)] × V/100 × W
where V = volume, W = fresh weigh, A663 = absorbance at 663 nm, and A645 = absorbance at 645 nm [23].

### 2.6. In Vitro Antifungal Test

A selected isolate of *Trichoderma* was tested in vitro for its ability to inhibit the growth of *R. solani*, the pathogen causing rice sheath blight, via a dual culture assay. Both *Trichoderma* and *R. solani* were cultured on PDA for 2 days before the bioassay. An agar plug of 3-day-old *R. solani* was placed on one side of the PDA in a Petri plate, and an agar plug of the 3-day-old Trichoderma isolate was placed on the opposite side, 5 cm away from *R. solani*. The PDA plate on which *R. solani* was cultured alone served as the control. The experiment was conducted via a complete randomized design (CRD) over three replicates, and the experiment was repeated twice. The tested Petri plates were incubated at an ambient temperature (28 ± 2 °C) until the pathogen covered the entire control plate. The colony radii of *R. solani* were measured and converted to a percentage of fungal growth inhibition via the following formula:Growth inhibition (%) = [(R1 − R2)/R1] × 100
where R1 = radial growth of *R. solani* in the control and R2 = radial growth of *R. solani* on the tested plate.

### 2.7. Siderophore Production Test

A test of the qualitative siderophore production of selected *Trichoderma* was carried out in chrome azurole agar (CAS) [24]. The CAS indicator (10 mL FeCl_3_, 50 mL CAS, and 40 mL CTAB) was dissolved in King’s B media containing 10 g glycerol, 1.5 g K_2_HPO_4_, 1.5 g MgSO_4_, and 20 g agar in 1 L DW [25]. The selected *Trichoderma* was cultured on a Petri dish containing CAS agar for 3 days. The experiment was conducted over three replicates and was repeated twice. The development of a clear zone and/or the discoloration of the agar from blue to yellow both indicated the production of siderophores.

### 2.8. Induction of Defense Responses in Rice by Selected Trichoderma Isolate

*Trichoderma*-treated and untreated rice seedlings, as described in Section 2.5, were subjected to analyses to determine the production of defense-related enzymes. Rice shoots and roots were collected and subjected to an enzyme assay. Rice seedlings were homogenized with phosphate buffer to pH 6.0 for peroxidase (POD) activity and to pH 7.5 for polyphenol oxidase (PPO) activity. The samples were then centrifuged at 14,000× *g* rpm at 4 °C for 10 min, and the supernatants were selected.

Peroxidase activity (POD) was assessed using the method of Nagle and Haard [26]. Here, 1% O-phenylenediamine (OPDA) was used as the substrate, which was combined with 0.3% H_2_O_2_. Reaction mixtures were measured using a UV-5300 UV/VIS spectrophotometer (METASH, Shanghai, China) at 420 nm, with the findings being expressed as U/mL. The polyphenol oxidase (PPO) assessment was performed via the method of Luh and Phithakpol [27]. Catechol was used as the substrate for the PPO activity assessment. The reaction mixtures were measured with a UV-5300 UV/VIS spectrophotometer (METASH, Shanghai, China) at 495 nm, and the results are expressed as U/mL.

### 2.9. Biological Control of Rhizoctonia solani in Pot Experiment

The conidial suspension was prepared as described in Section 2.5, and its concentration was adjusted to ×10^6^ conidia/mL. *Rhizoctonia solani* was cultured on PDA and incubated at an ambient temperature (28 ± 2 °C) for 7 days until sclerotia were observed. Sclerotia of *R. solani* were used as the inoculum. Rice seedlings were prepared according to the method described in Section 2.5. Rice seedlings were grown in a plastic pot (18 × 28 × 40 cm) containing 300 g sterilized silt clay soil, with one seedling per plastic pot. Rice seedlings were incubated in a greenhouse at a temperature ranging from 28 to 32 °C with natural light and were watered once a day. After 14 days of growth, the rice plants were subjected to experimentation. There were 4 treatments, comprising (i) healthy rice, (ii) pathogen inoculation, (iii) spraying with the conidial suspension 24 h before sclerotia inoculation, and (iv) sclerotia inoculation 24 h before spraying with the conidial suspension of *Trichoderma*. A total of 50 mL of the conidial suspension of *Trichoderma* was directly sprayed onto rice plants once a week. For pathogen inoculation, 3 sclerotia of *R. solani* were directly placed on the basal stems of the rice. The experiment was conducted over five replicates and was repeated twice. Then, rice plants were incubated at an ambient temperature for 30 days, and the plant growth was assessed using metrics such as the number of tillers and the above-ground height. Visual scoring was performed using the sheath blight incidence rating scale, following the method previously described by IRRI [28], converted to a percentage of disease incidence (PDI) using the following formula:$$\mathrm{PDI}=(\frac{Sum\;of\;individual\;rating}{No\;of\;leave\;examined\times Maximum\;disease\;scale})\times 100$$

### 2.10. Identification of Selected Trichoderma Isolate

The selected *Trichoderma* isolate was cultured on cornmeal dextrose agar, PDA, and synthetic nutrient agar (SNA), and incubated at an ambient temperature for 7 days so as to observe fungal growth [3]. The growth and colony characteristics of *Trichoderma* were observed and measured. Macroscopic and microscopic features were observed using a Leica S8AP0 stereomicroscope and Leica DM750 compound microscope (Leica Microsystems, Wetzlar, Germany). The dimensions of the conidiophores, phialides, and conidia were measured and photographed.

Genomic DNA was extracted from fungus cultured on PDA for five days using the FavoPrep^®^ DNA Extraction Mini Kit (Favorgen Biotech Corporation, Pingtung, Taiwan). The internal transcribed spacers (ITS), the second largest subunit of RNA polymerase II (*RPB*2), and the translation elongation factor 1-alpha (*TEF1-α*) genes were amplified with the primers ITS4/ITS5 [29], fRPB2-5f/fRPB2-7cr [30], and EF1-728F/EF1-986R [31], respectively. The amplification program ran over 35 cycles for all gene regions. The initial denaturation was induced at 95 °C for 5 min, followed by denaturation at 95 °C for 30 s, annealing at 52 °C for 30 s (ITS), 54 °C for 45 s (*rpb2*), and 52 °C for 1 min (*tef1-α*), extension at 72 °C for 1 min, and a final cycle at 72 °C for 10 min. The amplified PCR products were checked via the use of of 1% agarose gel electrophoresis, and we measured their quantity using NanoDrop OneC equipment (Thermo Scientific, Waltham, MA, USA). Then, the PCR products were purified using NucleoSpin Gel and a PCR Clean-up Kit (Macherey–Nagel, Germany), before being sent to a commercial sequencing provider (1^ST^ BASE Company, Kembangan, Malaysia). The obtained sequences were subjected to BLASTn searching in GenBank (http://blast.ddbj.nig.ac.jp/top-e.html, accessed on 25 March 2024).

The sequences derived in this study and from the GenBank database were used. Multiple sequence alignment was carried out using MUSCLE [32] and adjusted manually in BioEdit v.6.0.7 [33]. A phylogenetic tree was constructed using maximum likelihood (ML) on RAxML v7.0.3 in the CIPRES web portal under a GTRCAT model of nucleotide substitution with 1000 bootstraps [34,35]. The tree topologies were visualized using FigTree v1.4.0 [36].

### 2.11. Statistical Analysis

The results regarding IAA production, plant growth, and enzyme assays were subjected to one-way analysis of variance (ANOVA). Statistically significant differences between treatments were determined using Student’s *t*-test, and those among the treated samples were determined using Tukey’s test.

## 3. Results

### 3.1. Indole Compound-Producing Trichoderma Isolates and HPLC Analysis of Fungal IAA

A total of 21 *Trichoderma* isolates were cultured on PDB supplemented with 0.2% L-tryptophan, and the cultured broth was tested with the Salkowski reagent. The development of pink coloration in the *Trichoderma* culture broth indicated the production of indole compounds. The results show that only Z2-03 reacted positively to the Salkowski reagent. Therefore, we selected the *Trichoderma* Z2-03 isolate to confirm the production of IAA via HPLC, as well as for further bioassays on its ability to promote plant growth in rice and its antifungal capacity against *R. solani.*

### 3.2. HPLC Analysis of Fungal IAA

HPLC analysis was performed to identify and quantify the fungal IAA of crude metabolites extracted from the culture filtrates of *Trichoderma* sp. Z2-03. The HPLC chromatogram of crude extracts from *Trichoderma* isolate Z2-03 indicates the fungal IAA that corresponds to the IAA standard, with a retention time of 21.8 min (Figure 1) and a maximum absorption at 277 nm. The results indicate that isolate Z2-03 produced IAA at 35.58 ± 7.60 μg/mL.

### 3.3. Effect of Cell Free-CF on Seed Germination

The effect of the *Trichoderma* isolate Z2-03 cell-free CF on the induction of plant growth was measured on germinated rice at 4 dpa. The application of the selected *Trichoderma* isolate Z2-03 cell-free CF led to root and shoot lengths that were significantly greater than those in the control group (Figure 2, Appendix A). At 4 dpa, the root lengths of the DW, PDB, and cell-free CF-treated seeds were 2.83 ± 0.48, 2.69 ± 0.43, and 3.84 ± 0.35 cm, respectively, whereas the shoot lengths were 1.14 ± 0.14, 1.15 ± 0.18, and 1.98 ± 0.33 cm, respectively (*p* < 0.05). At 5 dpa, both the root and the shoot showed continuous germination; the root lengths of the DW, PDB, and cell-free CF-treated seeds were 2.90 ± 0.49, 2.77 ± 0.40, and 3.91 ± 0.43 cm, respectively, whereas the shoot lengths of DW, PDB, and cell-free CF-treated seeds were 1.51 ± 0.19, 1.40 ± 0.24, and 2.46 ± 0.45 cm, respectively (*p* < 0.05) (Figure 2, Appendix A).

### 3.4. Increase in Rice Seedling Growth Caused by Trichoderma Isolate Z2-03

In this study, we tested the effect of *Trichoderma* isolate Z2-03 on rice growth during the seedling stage; a spore suspension at a concentration of 10^6^ conidia/mL was applied to the rice seedlings, and the plant growth was measured at 10 days post-application using the shoot and root length and the fresh weights of the shoots and roots. The results show that the shoot and root lengths and the biomass of rice seedlings treated with *Trichoderma* isolate Z2-03 were significantly greater than those of the control (*p* < 0.05). The shoot length of rice seedlings in the control and treatment groups were 12.2 ± 1.22 and 15.30 ± 1.20 cm, respectively, whereas the root lengths of rice seedlings in the control and treatment groups were 8.85 ± 2.01 and 12.74 ± 1.97 cm, respectively (Figure 3). The fresh weights of the rice shoots were 0.196 ± 0.006 and 0.286 ± 0.015 mg per five seedlings for the *Trichoderma* isolate Z2-03-treated group and the control group, respectively. The root weights showed the same patterns, and the shoot weights were 0.050 ± 0.003 and 0.131 ± 0.002 mg per five seedlings, respectively (Figure 3, Appendix A). Furthermore, the total chlorophyll contents of the *Trichoderma* isolate Z2-03-treated group and the group were measured; the former was 5.093 ± 0.054 mg, which is significantly higher than that of the control, at 1.352 ± 0.001 mg (Figure 3, Appendix A).

### 3.5. Antifungal Ability of Trichoderma Isolate Z2-03

In order to test the antifungal ability of the IAA-produced *Trichoderma* isolates, the selected *Trichoderma* isolate Z2-03 was tested against the fungal growth of *R. solani* using a dual culture assay. At 7 days of incubation, the mycelial radius of *R. solani* on the control plate was 6 cm, which is significantly higher than that on the plate tested with *Trichoderma* isolate Z2-03 (Figure 4A–E, Appendix A). The results show that Z2-03 suppressed the mycelial growth of *R. solani*, with a percentage of inhibition of 74.17%. Furthermore, stereo microscopy was used to elucidate the conidia, mycelia, and colony pustule of Z2-03 colonized on a colony of *R. solani* in the test group (Figure 4F,G).

### 3.6. Siderophore Production of Trichoderma Isolate Z2-03

Qualitative siderophore production was carried out on a Petri dish containing CAS media, and the change in the color of the media from blue to yellow/orange indicated the production of siderophore. After culturing the *Trichoderma* isolate Z2-03 on CAS media for 1–7 days, the first color change in CAS media occurred on the *Trichoderma* isolate Z2-03 colony at day 3, a clear yellow zone developed around the colony of *Trichoderma* isolate Z2-03 at day 5, and full discoloration from blue to yellow occurred on day 7 with incubation at an ambient temperature (Figure 5, Appendix A). These results reveal the production of siderophore in the selected *Trichoderma* isolate Z2-03.

### 3.7. Induction of Defense Responses in Rice by Trichoderma Isolate Z2-03

In order to investigate the plant defense response in rice after treatment with *Trichoderma* isolate Z2-03, the activities of plant defense-related enzymes such as peroxidase (POD) and polyphenol oxidase (PPO) were assayed. Our results show that the POD activity in *Trichoderma* isolate Z2-03-treated shoots was 490.68 ± 34.23, which is significantly greater than that in the control, at 129.90 ± 25.88 unit per mL (U/mL) (Figure 6). The POD activity of *Trichoderma* isolate Z2-03-treated roots was 530.69 ± 2.08, which is significantly lower than that of the control, at 641.40 ± 28.18 U/mL (Figure 6). Furthermore, the PPO activity of the *Trichoderma* isolate Z2-03-treated shoots was 30.74 ± 2.05 U/mL, which is significantly lower than that of the control, at 44.29 ± 4.40 U/mL (Figure 6). On the other hand, the PPO activity of *Trichoderma* isolate Z2-03-treated roots was 27.15 ± 1.69 U/mL, which is significantly higher than that of the control (Figure 6).

### 3.8. Trichoderma Promote Plant Growth and Reduce Disease Incidence

After incubation for 30 days, the application of the *Trichoderma* isolate’s Z2-03 conidial suspension before inoculation with the *R. solani* treatment (T1) yielded 3.63 tillers, which is significantly higher than the numbers in other treatment (Figure 7A). Regarding the plant height, the application of the *Trichoderma* isolate Z2-03 conidial suspension yielded a height of 63.50 cm, which is significantly greater than the heights measured in other treatments (Figure 7B). Furthermore, the application of the *Trichoderma* isolate Z2-03 conidial suspension led to a PDI of 67.85%, which is significantly lower than those measured in other treatments (Figure 7C, Appendix A).

### 3.9. Identification of Trichoderma Isolate Z2-03

*Trichoderma* isolate Z2-03 grew to cover Petri dishes containing CMD, PDA, and SNA within 3, 4, and 4 days, respectively. The white cotton colony of *Trichoderma* isolate Z2-03 was grown on CMD and PDA; it produced green to greenish conidia at the center of the colony, whereas, on SNA, it grew tufts of conidia at 4 days of incubation (Figure 8). The conidiophore showed pyramidal-type branches. Phialides were of the ampulliform type, measuring 5.66–13.70 μm long and 2.82–5.24 μm wide (average ± SD, 8.83 ± 2.0 × 3.68 ± 0.54, *n* = 20). Conidia were light green to greenish, globose to subglobose, and were 2.52–3.63 μm long and 2.49–3.26 μm wide (3.08 ± 0.32 × 2.73 ± 0.26, *n* = 20). The chlamydospore were of globose to subglobose form, and were 5.20–7.84 μm long and 4.98–7.34 μm wide (6.43 ± 0.69 × 6.10 ± 0.78, *n* = 20). The morphological characterization revealed this isolate to be *Trichoderma* sp.

The ITS, *rbp*2, and *tef1-α* sequences of isolate Z2-03 were deposited in GenBank as PP528686, PP539902, and PP539901, respectively. The combined ITS, *rbp*2, and *tef1-α* sequence dataset consisted of 23 taxa, and the aligned dataset comprised 1990 characters including gaps (ITS: 1–575; *rbp*2: 576–1725; *tef1-α*: 1726–1990). *Trichroderma hunamense* HMAS24841 and *T. longisporum* HMAS24843 were set as outgroups. The sequence alignment yielded 396 distinct alignment patterns, with 9.19% undetermined characters or gaps. The RAxML analysis resulted in a final ML optimization likelihood value of –5978.0906. The phylogenetic tree is shown in Figure 9. The results indicate that the fungal isolate Z2-03 obtained in this study was successfully assigned to the same clade as *Trichoderma breve*, containing the species (HMAS 248844). This clade established a monophyletic clade with a 100% BS support value, and it formed a sister taxon to *T. zelobreve* with a 100% BS support value. Therefore, the fungal isolate Z2-03 was identified as *T*. *breve*.

## 4. Discussion

In this study, several isolates of *Trichoderma* were screened for IAA production, and *Trichoderma* isolate Z2-03 was found to be the most effective in this respect. *Trichoderma* isolate Z2-03 was identified based on its morphology and molecular properties, as *T. breve* displayed a plant growth-promoting ability through increasing the root and shoot length, fresh weight, and total chlorophyll content in “Chor Khing” rice variety seedlings. Furthermore, *T. breve* isolate Z2-03 exhibited a fungicidal ability against *R. solani,* the pathogen causing sheath blight in “Chor Khing” rice variety seedlings, through competition, parasitism, and the induction of defense resistance.

The Salkowski reagent is known to react with indole compounds such as IAA, indoleacetamide (IAM), and indolepyruvic acid [37]. However, in this study, only *T. breve* Z2-03 tested positive for the Salkowski reagent, indicating the presence of IAA or other indole compounds. To quantify specific IAA, we conducted HPLC analysis using IAA standard. Our results confirmed the presence of microbial IAA in the cell-free culture filtrate of *T. breve* Z2-03 (Figure 1). This suggests that the detection of IAA initially relied on screening for indole compounds using the Salkowski reagent, followed by the quantification of specific IAA through HPLC analysis.

It has been determined that some micro-organisms are able to produce phytohormones like IAA [38,39]. For instance, *T. harzianum* and *T. virens* produce IAA with production levels that correlate with the amount of tryptophan present in the medium [40,41]. In this study, culturing *T. breve* isolate Z2-03 in a culture medium supplied with 2% tryptophan led to the production of IAA, as determined by the colorimetric method using the Salkowski reagent. *Trichoderma harzianum*, the IAA-producing strain, has been reported to control anthracnose disease and enhance growth in sorghum plants [40]. The in vitro bioassay showed that *T. virens* isolate Gv29.8 and *T. atroviride* isolate IMI206040 are able to synthesize IAA, which increases the lateral roots in *Arabidopsis* [40]. Furthermore, Mahmoodian et al. [42] showed that the inoculation of *T. harzianum* increased the biomass, seedling height, and root length, and induced resistance in bean plants against *Rhizoctonia*. Our result is in agreement with previous research, which found that the application of *T. breve* isolate Z2-03 increases the shoot and root length, as well as the biomass of shoots and roots in rice seedlings. Therefore, we can infer that *T. breve* isolate Z2-03 is involved in promoting plant growth in the rice variety “Chor Khing”.

It not only promotes plant growth, but a fungicidal ability is commonly observed in several strains of *Trichoderma* species. Due to its capacity to compete for nutrients and space, produce siderophores, redirect parasites, and induce defense responses in plant, *Trichoderma* species are widely applied to control numerous plant diseases. For instance, *T. gamsii* T6085 displayed competitiveness and root endophytism in controlling *Fusarium* head blight in wheat [13]. The *T. harzianum* isolate TRIC8 competed for seed colonization against seedborne pathogens *Alternaria alternata*, *Bipolaris cynodontis, Fusarium culmorum*, and *F*. *oxysporum* [43]. Siderophores are chelating agents with a high affinity for ferric iron. Siderophores play an important role in increasing the availability of iron through the solubilization of iron in a precipitated form [44]. Some micro-organisms are able to produce siderophores under iron-limited conditions [45]. The secretion of siderophore by micro-organisms elevated competitiveness through enabling the sequestering of available iron and making it unavailable to competitors. Using a dual culture assay and a qualitative siderophore test in the CAS medium, our study has demonstrated the competitiveness of *T. breve* isolate Z2-03 against *R. solani*. This finding indicates that the competitiveness of *T. breve* isolate Z2-03 may relate to its fungicidal ability against the sheath blight disease pathogen of the “Chor Khing” rice variety.

Plant responses can be stimulated by use of a biological control agent (BCA), which can induce defense-related enzymes such as peroxidase (POD) and polyphenol oxidase (PPO) in plants [14,46]. POD plays an important role in plant defense against pathogens, the biosynthesis of lignin [47], and the catalyzing of the oxidation of hydrogen peroxide (H_2_O_2_) during pathogen attacks. On the other hand, PPO activity is responsible for catalyzing the conversion of phenols into quinones and preventing serious damage to plants caused by pathogens during plant responses to disease [48]. A high level of POD activity during pathogen infection has been observed in different plant species, including cucumber [49], rice [50], and tomato [51]. Furthermore, the application of a talc-based formulation of *T. viride* and *Pseudomonas fluorescens* containing chitin increased POD and PPO activities, which are responsible for coconut palm’s antifungal ability against *Ganoderma* [52]. Our study is in agreement with previous research showing that high levels of POD and PPO activity were detected in the shoots and roots, respectively, of *T. breve* isolate Z2-03-treated rice seedlings. Our findings show that POD and PPO activities differed in different parts of the plant; this may due to the expression levels of POD and PPO in the shoots and roots of plants, but we did not observe this aspect in this study.

The *Trichoderma* species has been widely applied for disease control and to promote plant growth in pot experiments and field trials. Our study has revealed that the application of the *T. breve* isolate Z2-03 conidial suspension increased the number of tillers and reduced the PDI in comparison with other treatments. Our findings are in agreement with those of Doni et al. [53], who found that inoculation with the *Trichoderma* isolate SL2 increased the number of tillers and the plant height in rice. It has been shown that inoculation with the *Trichoderma* strain protects against *R. solani* by reducing the susceptibility index in *Oryza sativa* L. Malaysian cultivar MRQ74 [54]. In this study, the inoculation of *T. breve* isolate Z2-03 reduced the disease severity by reducing the percentage of disease incidence (PDI), which could ultimately help protect the plant from *R. solani*.

*Trichoderma breve* Z2-03 showed that it produced and released IAA and siderophore. IAA is involved in plant growth and development, whereas siderophores are responsible for competing with the sheath blight pathogen *R. solani*. IAA is one of the auxin hormones responsible for cell division and cell elongation in plants, has been recognized to be involved in plant growth and development, and is part of the complex network of plant–pathogen interactions. For instance, the overexpression of the IAA *MdIAA24* gene has been reported to enhance *Glomerella* leaf spot resistance in apples via improved reactive oxygen species (ROS) scavenging and defense-related enzyme activity [55]. The application of IAA has been shown to reduce the spore germination, mycelial dry weight, and protein content of tomato wilt pathogen *Fusarium oxysporum* f.sp. *lycopersici,* as well as improve the growth and yield of the shoot and root of tomato plants, which induces resistance in tomato plants against *F*. *oxysporum* f.sp. *lycopersici* [56]. Moreover, siderophores have been reported to trigger plant immunity and act as part of the induced systemic resistance (ISR). For instance, the non-plant pathogenic bacterium *Pseudomonas aeruginosa* 7NSK2 produces two types of siderophores, namely pyoverdine and pyochelin [57]. It has been shown that pyochelin is involved in protecting against the plant pathogen oomycete *Pythium splendens*, which causes damping-off in tomatoes. Therefore, the production of IAA and siderophore by *T. breve* Z2-03 in this study may be associated with disease resistance against sheath blight caused by *R. solani*.

The *Trichoderma breve* type strain HMAS24844 was first isolated from a soil sample in China and identified based on the morphology and molecular properties of *tef1-α* and *rpb2* genes [58]. The application of *T. breve* has shown the highest chlorophyll index and increased the plant height and leaf number of *Theobroma cacao* L. [59]. Furthermore, *T. breve* isolated from vermicompost effectively inhibited the fungal growth of *Colletotrichum siamense*, which causes anthracnose on chili [60]. Our findings reveal that *T. breve* Z2-03 not only promotes plant growth, but also controls *R. solani*, the pathogen of rice sheath blight in the “Chor Khing” variety of rice. Although the biological control of *Trichoderma* is widely used to suppress plant diseases, the use of fresh *Trichoderma* conidia in the field may be affected by environmental conditions such as high temperature, light, rain, and the shelf life of the formulation. The development of appropriate formulations, such as emulsion formulation, can support a long shelf life and be ready-to-use with a high concentration of conidia and a high dilution ratio with water [61]. However, we did not develop the appropriate formulation or evaluate cost-effectiveness in this study. To develop the appropriate formulation, testing its ability and evaluating cost-effectiveness is still necessary to verify our findings in the near future.

## 5. Conclusions

In this study, soil fungus was screened for IAA productivity, its ability to increase plant growth in the “Chor Khing” rice variety, and its fungicidal activity against *R. solani*—the pathogen causing sheath blight in rice. The most effective fungus was tentatively identified based on morphology and molecular data as *T. breve* isolate Z2-03. *Trichoderma breve* isolate Z2-03 was shown to induce the production of IAA and increase the shoot and root length, fresh weight, as well as total chlorophyll content in the tested rice plants. Furthermore, *T. breve* isolate Z2-03 was characterized by its competitiveness and its capacity to induce a defense response against *R. solani*. The present study reveals that *T. breve* isolate Z2-03 has the potential to be used as a biocontrol agent and plant growth-promoting fungus. However, the appropriate concentration of *T. breve* isolate Z2-03 for use in the field and in field trials needs to be verified in the future.

## Figures and Tables

**Figure 1 jof-10-00417-f001:**
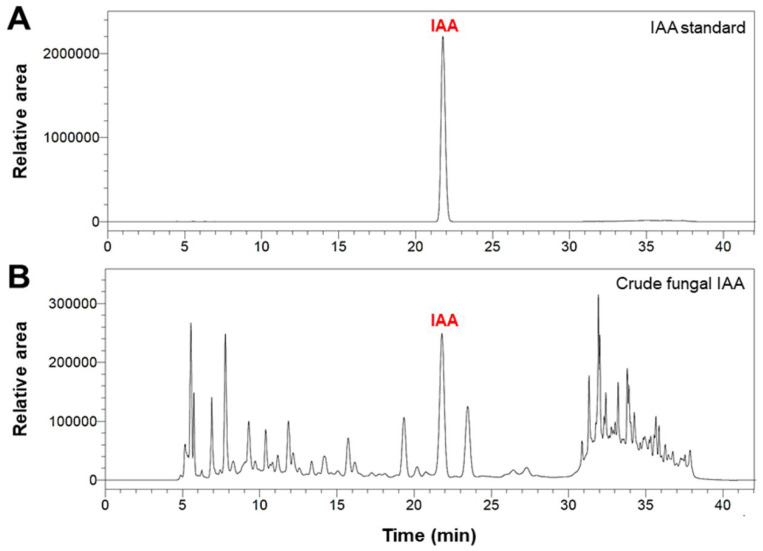
HPLC chromatograms of IAA standard at a concentration of 1.0 mg/mL (**A**) and crude fungal IAA (**B**).

**Figure 2 jof-10-00417-f002:**
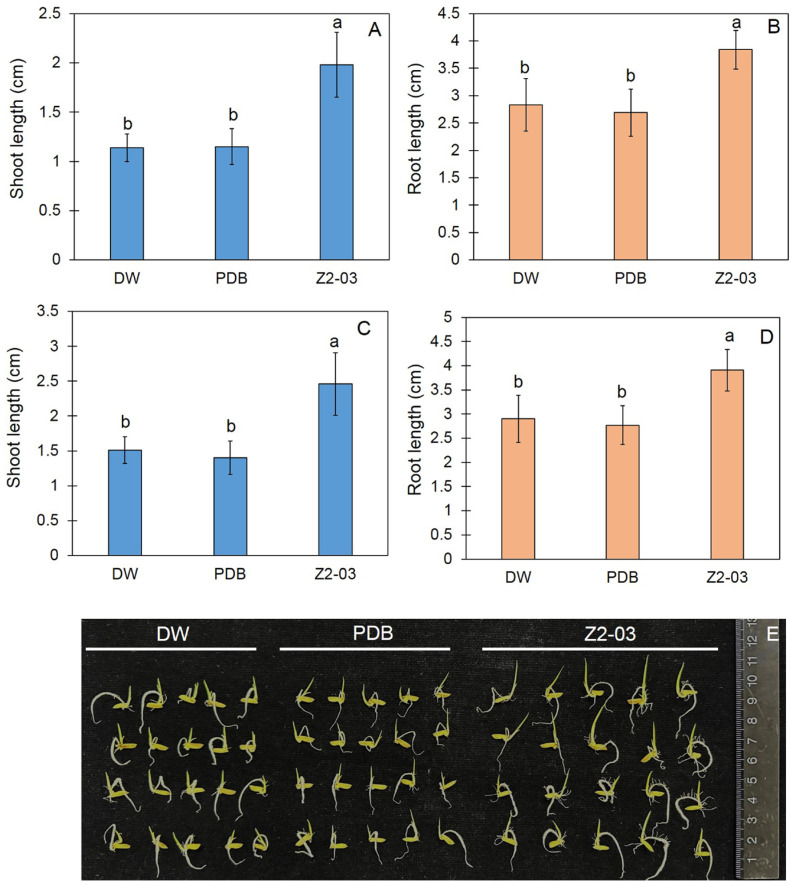
Seed germination, as well as the shoot and root length, of the “Chor Khing” rice variety assessed using distilled water (DW), potato dextrose agar broth (PDB), and the cell-free culture filtrate of *Trichoderma* isolate Z2-03 (Z2-03). Shoot and root length (**A**,**B**) at 4 days post application (dpa) and at 5 dpa (**C**,**D**), as well as the phenotypes of the germinated shoot and root at 4 dpa (**E**). Values are shown as mean ± SD, with letters indicating significant differences among the control and the treatment according to Tukey’s test (*p* < 0.05).

**Figure 3 jof-10-00417-f003:**
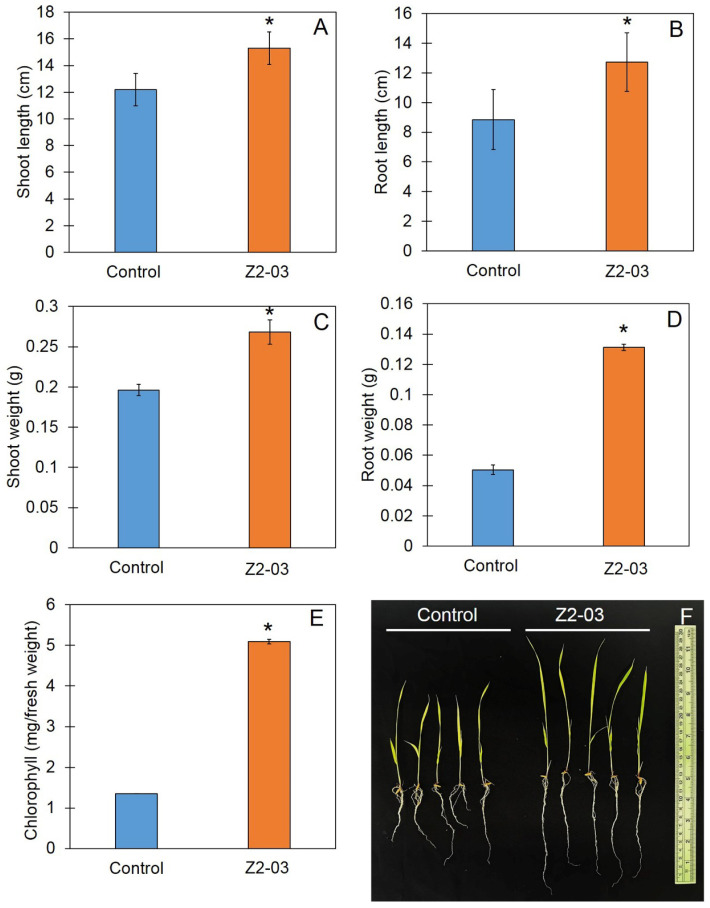
Shoot and root length and biomass of the “Chor Khing” rice variety assessed via the distilled water (control) and spore suspension of *Trichoderma* isolate Z2-03. Shoot and root length (**A**,**B**), fresh weights of the shoot (**C**) and root (**D**), total chlorophyll content (**E**), and phenotypes of rice seedlings (**F**). Values are mean ± SD; asterisks indicate significant difference between control and treatment groups according to Student’s *t*-test (*p* < 0.05).

**Figure 4 jof-10-00417-f004:**
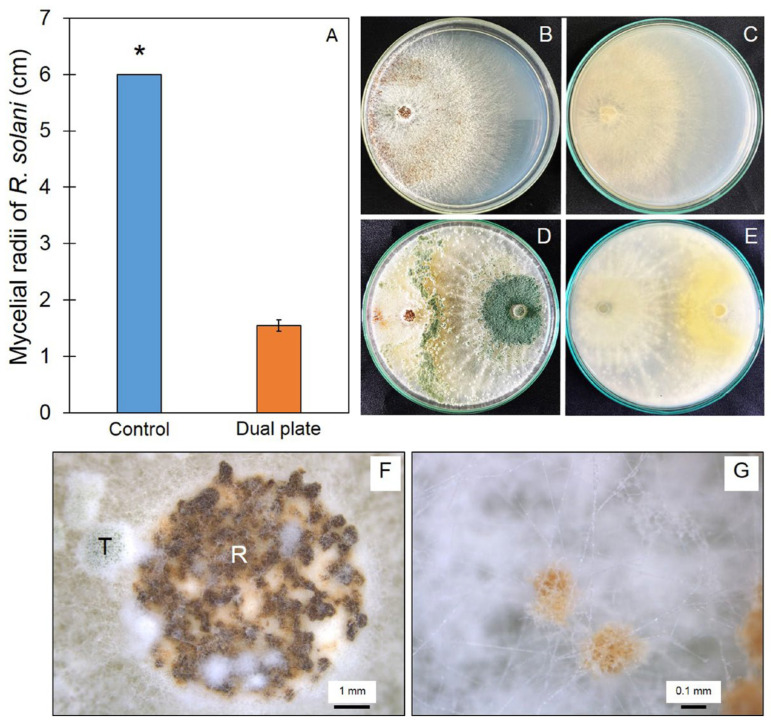
Antagonistic activity of *Trichoderma* isolate Z2-03 against *Rhizoctonia solani*. Mycelial radii of *R. solani* in control and dual culture plates (**A**), growth of *R. solani* from the top (**B**) and bottom view (**C**), *R. solani* on the dual culture plate from the top (**D**) and bottom view (**E**), colonization of Trichoderma mycelia and conidia on sclerotia of *R. solani* (**F**,**G**). Values are mean ± SD, asterisks indicate a significant difference between the control and treatment according to Student’s *t*-test (*p* < 0.05). T indicates *Trichoderma* Z2-03 and R indicates *R. solani*.

**Figure 5 jof-10-00417-f005:**
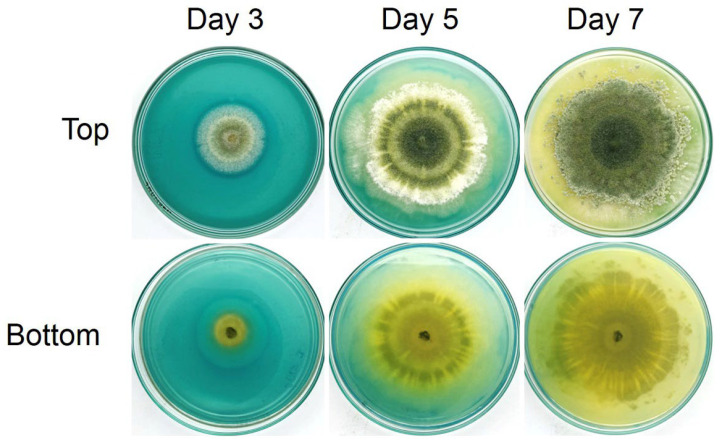
Siderophore production of *Trichoderma* isolate Z2-03 on the CAS medium incubated at an ambient temperature at 3, 5, and 7 days post-incubation. The development of a yellow clear zone on the CAS medium indicates siderophore production.

**Figure 6 jof-10-00417-f006:**
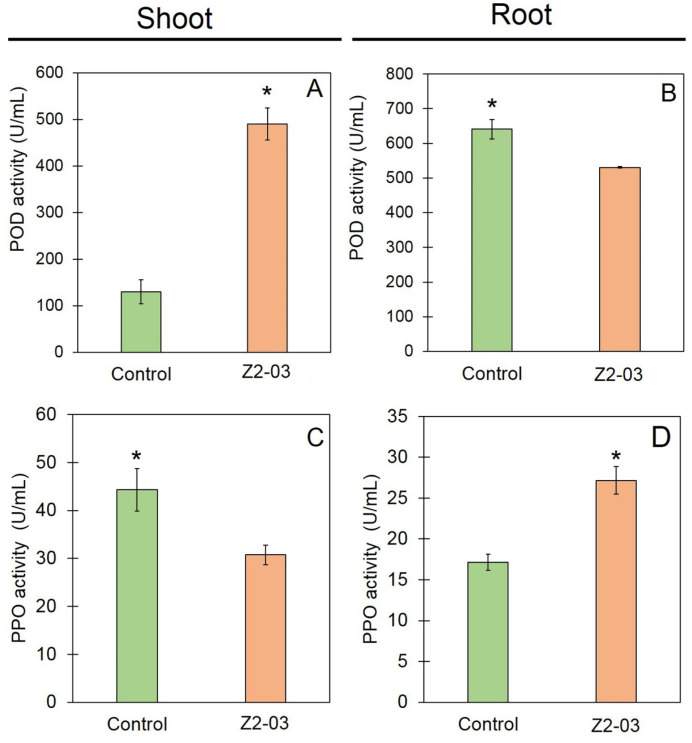
Enzyme activities of defense-related enzymes in *Trichoderma* isolate Z2-03-treated rice seedlings and the control group. Peroxidase (POD) activity in the shoot (**A**) and root (**B**), and polyphenol oxidase (PPO) activity in the shoot (**C**) and root (**D**). Values are mean ± SD; asterisks indicate a significant difference between the control and treatment groups according to Student’s *t*-test (*p* < 0.05).

**Figure 7 jof-10-00417-f007:**
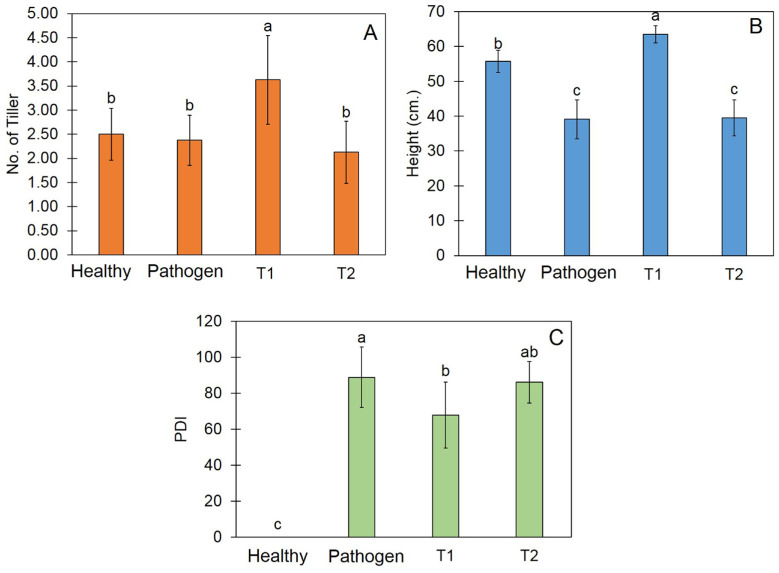
Effect of the conidial suspension of *Trichoderma* isolate Z2-03 on the number of tillers (**A**), plant height (**B**), and percentage of disease incidence (**C**). Values are means ± SD, and letters indicate a significant difference among the treatments according to Tukey’s test (*p* < 0.05). T1 indicates the application of *Trichoderma* isolate Z2-03 prior to *Rhizoctonia solani* inoculation, whereas T2 indicates inoculation with *R. solani* prior to *Trichoderma* isolate Z2-03 application.

**Figure 8 jof-10-00417-f008:**
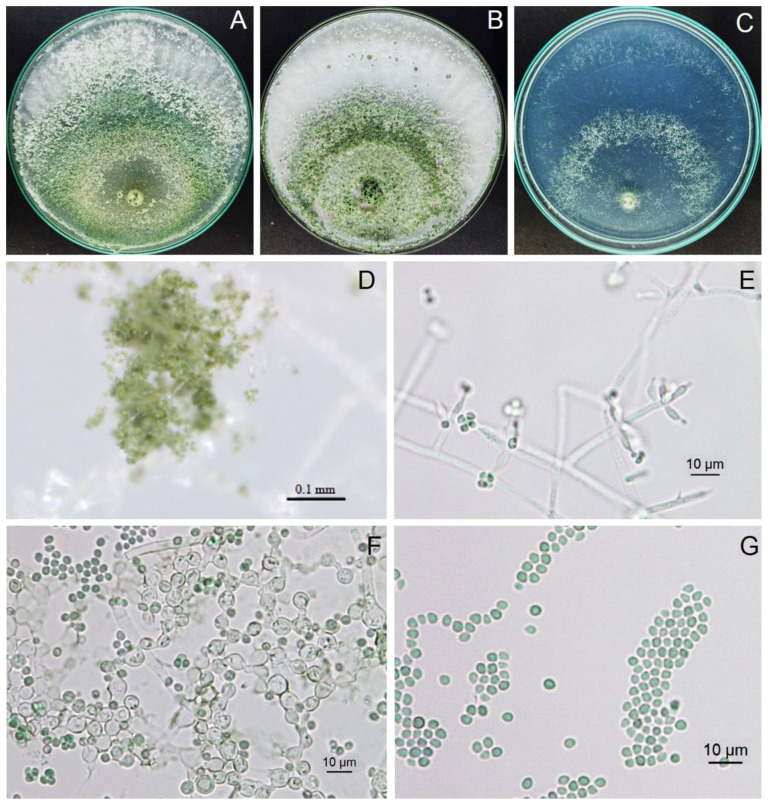
Morphology of *Trichoderma breve* isolate Z2-03, colonized on CMD (**A**), PDA (**B**), and SNA (**C**); floccose mat on PDA (**D**), conidiophores and phialides (**E**), chlamydospores (**F**), and conidia (**G**).

**Figure 9 jof-10-00417-f009:**
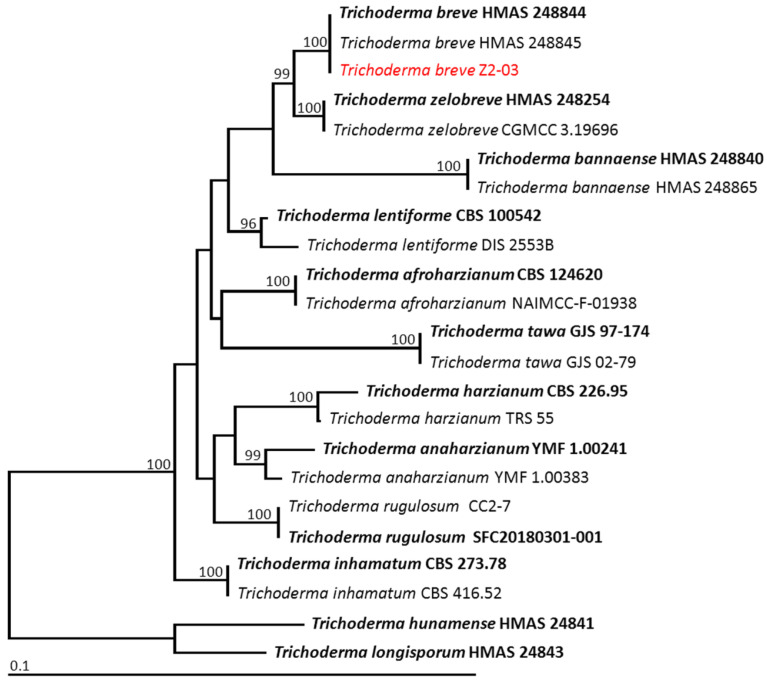
Phylogram derived from the maximum likelihood analysis of 23 fungal isolates of the combined ITS, *rpb2*, and *tef1-α*. *Trichroderma hunamense* HMAS24841 and *T. longisporum* HMAS24843 were set as the outgroup. The numbers above the branches represent bootstrap percentages, and values > 95% are shown. The scale bar represents the expected number of nucleotide substitutions per site. The fungal isolates obtained from this study are shown in red. Type species are shown in bold.

## Data Availability

The DNA sequence data obtained from this study were deposited in GenBank under accession numbers ITS (PP528686), *rpb2* (PP539902), and *tef1-α* (PP539901).

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
