# Peer review of "Plant Growth Promotion and Biological Control against Rhizoctonia solani in Thai Local Rice Variety “Chor Khing” Using Trichoderma breve Z2-03"

_jof, 2024, doi:10.3390/jof10060417_

Round 1
Reviewer 1 Report
The manuscript titled "Plant Growth Promotion and Biological Control Against Rhizoctonia solani in Thai Local Rice Variety 'Chor Khing' using Trichoderma breve Z2-03" presents findings on the dual role of Trichoderma breve Z2-03 as a plant growth promoter and a biological control agent in rice. The study presents potential findings that could benefit agricultural practices and ecological management. However, the manuscript requires major revisions to enhance its scientific rigor, clarity, and contextual depth. Once these issues are addressed, the manuscript would make a valuable contribution to the field of agricultural microbiology.
Line 108: Do not start any sentence with abbreviation
Scientific names must be italicized for example in line 166
More detail is required for pot experiment. For example how much soil was used per pot?
What was the composition of soil used in the experiment? Sterilized soil or field soil?
How did you maintain only one seedling in a single pot?
The experiment was repeated twice, however, I did not find the data of repeated experiment
Line 288: there is no different letters affiliated with treatment values
For antifungal activity test, a positive control should be used
There is no direct data on pathogen suppression during pot experiment
The use of controls in the experiments should be specified more clearly. For instance, it is mentioned that the control in the germination test was "culture broth alone." It would be useful to know if this broth was autoclaved or treated to ensure it did not contain any Trichoderma or other microbial contaminants.
The paper notes that the Salkowski reagent was used to detect indole compounds, and specifically IAA was quantified. It should be noted that the Salkowski reagent is not specific to IAA and can react with other indoles. The authors should discuss the possibility of other indole compounds influencing their results.
While the paper discusses the production of IAA and siderophores as mechanisms of growth promotion and disease suppression, it would strengthen the study to include more direct evidence linking these mechanisms to the observed effects on plant growth and disease resistance.
The molecular identification of T. breve Z2-03 is based on ITS, tef1-, and rpb2 sequences. It would enhance the credibility of this identification if the authors provided accession numbers for these sequences if they are deposited in a public database, allowing other researchers to verify the identification.
The discussion could be expanded to address practical aspects of applying Trichoderma breve Z2-03 in agricultural settings, such as formulation, delivery methods, and cost-effectiveness, which are critical for end-users considering this biological control method.
These comments aim to help refine the manuscript and potentially increase its impact in the field of plant pathology and agronomy.
The manuscript would benefit significantly from improvements in English language usage to enhance clarity and readability. Careful editing to correct grammatical errors, ensure consistent terminology, and improve sentence structure will help in conveying the scientific findings more effectively. It is recommended to engage a native English speaker or a professional editor with a science background to ensure that the language meets academic standards.
Line 108: Do not start any sentence with abbreviation
Scientific names must be italicized for example in line 166
More detail is required for pot experiment. For example how much soil was used per pot?
What was the composition of soil used in the experiment? Sterilized soil or field soil?
How did you maintain only one seedling in a single pot?
The experiment was repeated twice, however, I did not find the data of repeated experiment
Line 288: there is no different letters affiliated with treatment values
For antifungal activity test, a positive control should be used
There is no direct data on pathogen suppression during pot experiment
Author Response
Major comments
The manuscript titled "Plant Growth Promotion and Biological Control Against Rhizoctonia solani in Thai Local Rice Variety 'Chor Khing' using Trichoderma breve Z2-03" presents findings on the dual role of Trichoderma breve Z2-03 as a plant growth promoter and a biological control agent in rice. The study presents potential findings that could benefit agricultural practices and ecological management. However, the manuscript requires major revisions to enhance its scientific rigor, clarity, and contextual depth. Once these issues are addressed, the manuscript would make a valuable contribution to the field of agricultural microbiology.
Answer: Thank you for reviewing the manuscript and providing valuable comments to enhance its quality.
Line 108: Do not start any sentence with abbreviation
Answer: We have revised "Indole-3-acetic acid" to be used instead of "IAA" at the beginning of this sentence.
Scientific names must be italicized for example in line 166
Answer: We have italicized “Trichoderma” in line 166.
More detail is required for pot experiment. For example how much soil was used per pot?
Answer: We have added 300 g of sterilized soil.
What was the composition of soil used in the experiment? Sterilized soil or field soil?
Answer: In this study, we used sterilized soil to prevent contamination from field soil.
How did you maintain only one seedling in a single pot?
Answer: We have added “Rice seedlings were incubated in a greenhouse at a temperature ranging from 28 to 32°C, with natural light and watered once a day.” into materials and methods.
The experiment was repeated twice, however, I did not find the data of repeated experiment
Answer: Actually, both sets of data were calculated as averages and standard deviations. However, both the initial and repeated data showed similar results. Therefore, we only presented the single set of data in this manuscript.
Line 288: there is no different letters affiliated with treatment values
Answer: The figure has been revised, and we have added letters to the graph.
For antifungal activity test, a positive control should be used
Answer: In this study, we used the "dual culture assay" to test the antifungal ability of Trichoderma Z2-03. The experiment included the pathogen alone as a control, and a dual culture tested plate containing Trichoderma and the pathogen, as previously reported by several researchers.
There is no direct data on pathogen suppression during pot experiment
Answer: In this pot experiment, it is difficult to directly evaluate pathogen suppression as observed in vitro. Instead of assessing direct pathogen suppression, we applied Trichoderma to rice and measured the percentage disease incidence (PDC) on the rice.
The use of controls in the experiments should be specified more clearly. For instance, it is mentioned that the control in the germination test was "culture broth alone." It would be useful to know if this broth was autoclaved or treated to ensure it did not contain any Trichoderma or other microbial contaminants.
Answer: In this experiment, we utilized autoclaved culture broth as the control. The text now includes the information that the "control group was soaked with autoclaved PDB alone."
The paper notes that the Salkowski reagent was used to detect indole compounds, and specifically IAA was quantified. It should be noted that the Salkowski reagent is not specific to IAA and can react with other indoles. The authors should discuss the possibility of other indole compounds influencing their results.
Answer: We have discussed about this phenomenon as “The Salkowski reagent is known to react with indole compounds such as IAA, indoleacetamide (IAM), and indolepyruvic acid [37]. However, in this study, only T. breve Z2-03 tested positive for Salkowski reagent, indicating the presence of IAA or other indole compounds. To quantify specific IAA, we conducted HPLC analysis using IAA standard. Our results confirmed the presence of microbial IAA in the cell-free culture filtrate of T. breve Z2-03 (Fig. 1). This suggests that the detection of IAA initially relied on screening for indole compounds using the Salkowski reagent, followed by the quantification of specific IAA through HPLC analysis.
While the paper discusses the production of IAA and siderophores as mechanisms of growth promotion and disease suppression, it would strengthen the study to include more direct evidence linking these mechanisms to the observed effects on plant growth and disease resistance.
Answer: We have added more discussion about this phenomenon as “Trichoderma breve Z2-03 showed that it produced and released IAA and siderophore. IAA is involved in plant growth and development, whereas siderophores are responsible for competing with the sheath blight pathogen R. solani. IAA is one of the auxin hormones responsible for cell division and cell elongation in plants, has been recognized to be involved in plant growth and development, and is part of the complex network of plant-pathogen interactions. For instance, overexpression of IAA MdIAA24 gene has been reported to enhance Glomerella leaf spot resistance in apples via improved reactive oxygen species (ROS) scavenging and defense-related enzyme activity [55]. Application of IAA has been shown to reduce spore germination, mycelial dry weight, and protein content of tomato wilt pathogen Fusarium oxysporum f.sp.lycopersici as well as improve growth and yield of shoot and root of tomato which leads to induce resistance in tomato plants against F. oxysporumf.sp. lycopersici [56]. Moreover, siderophores have been reported to trigger plant immunity and act as part of induced systemic resistance (ISR). For instance, the non-plant pathogenic bacterium Pseudomonas aeruginosa 7NSK2 produces two types of siderophores: pyoverdine and pyochelin [57]. It has been shown that pyochelin is involved in protecting against the plant pathogen oomycete Pythium splendens, which causes damping-off in tomatoes. Therefore, the production of IAA and siderophore by T. breve Z2-03 in this study may be associated with disease resistance against sheath blight caused by R. solani.
The molecular identification of T. breve Z2-03 is based on ITS, tef1-, and rpb2 sequences. It would enhance the credibility of this identification if the authors provided accession numbers for these sequences if they are deposited in a public database, allowing other researchers to verify the identification.
Answer: We have deposited DNA sequences of ITS, rpb2 and tef1-a in GenBank with accession number PP528686, PP539902, and PP539901, respectively.
The discussion could be expanded to address practical aspects of applying Trichoderma breve Z2-03 in agricultural settings, such as formulation, delivery methods, and cost-effectiveness, which are critical for end-users considering this biological control method.
Answer: We have added discussion as “The Trichoderma breve type strain HMAS24844 was first isolated from a soil sample in China and identified based on morphology and molecular properties of tef1-a and rpb2 genes [58]. The application of T. breve has shown the highest chlorophyll index and increased plant height and leaf number of Theobroma cacao L. [59]. Furthermore, T. breve isolated from vermicompost effectively inhibited fungal growth of Colletotrichum siamense, which causes anthracnose on chili [60]. Our findings reveal that T. breve Z2-03 not only promotes plant growth, but also controls R. solani, the pathogen of rice sheath blight in the "Chor Khing" variety of rice. Although biological control of Trichoderma is widely used to suppress plant diseases, the use of fresh Trichoderma conidia in the field may be affected by environmental conditions such as high temperature, light, rain, and the shelf life of the formulation. Development of appropriate formulations, such as emulsion formulation, can support a long shelf life and be ready-to-use with a high concentration of conidia and a high dilution ratio with water [61]. However, we did not develop the appropriate formulation or evaluate cost-effectiveness in this study. To develop the appropriate formulation, testing its ability and evaluating cost-effectiveness is still necessary to verify soon.
These comments aim to help refine the manuscript and potentially increase its impact in the field of plant pathology and agronomy.
Answer: We appreciate your valuable comments on improving the manuscript. Thank you again.
The manuscript would benefit significantly from improvements in English language usage to enhance clarity and readability. Careful editing to correct grammatical errors, ensure consistent terminology, and improve sentence structure will help in conveying the scientific findings more effectively. It is recommended to engage a native English speaker or a professional editor with a science background to ensure that the language meets academic standards.
Answer: Thank you for your concern regarding the academic English used in this study. This manuscript has been proofread and edited by the MDPI English editing service.
Detail comments
Line 108: Do not start any sentence with abbreviation
Answer: We have revised "Indole-3-acetic acid" to be used instead of "IAA" at the beginning of this sentence.
Scientific names must be italicized for example in line 166
Answer: We have italicized “Trichoderma” in line 166.
More detail is required for pot experiment. For example how much soil was used per pot?
Answer: We have added 300 g of sterilized soil.
What was the composition of soil used in the experiment? Sterilized soil or field soil?
Answer: In this study, we used sterilized soil to prevent contamination from field soil.
How did you maintain only one seedling in a single pot?
Answer: We have added “Rice seedlings were incubated in a greenhouse at a temperature ranging from 28 to 32°C, with natural light and watered once a day.” into materials and methods.
The experiment was repeated twice, however, I did not find the data of repeated experiment
Answer: Actually, both sets of data were calculated as averages and standard deviations. However, both the initial and repeated data showed similar results. Therefore, we only presented the single set of data in this manuscript.
Line 288: there is no different letters affiliated with treatment values
Answer: The figure has been revised, and we have added letters to the graph.
For antifungal activity test, a positive control should be used
Answer: In this study, we used the "dual culture assay" to test the antifungal ability of Trichoderma Z2-03. The experiment included the pathogen alone as a control, and a dual culture tested plate containing Trichoderma and the pathogen, as previously reported by several researchers.
There is no direct data on pathogen suppression during pot experiment
Answer: In this pot experiment, it is difficult to directly evaluate pathogen suppression as observed in vitro. Instead of assessing direct pathogen suppression, we applied Trichoderma to rice and measured the percentage disease incidence (PDC) on the rice.
Reviewer 2 Report
A reviewer opinion on manuscript entitled “ Plant Growth Promotion and Biological Control Against Rhizoctonia solani in Thai Local Rice Variety “Chor Khing” using Trichoderma breve Z2-03 “ by Warin Intana et al.
In my opinion this is interesting case study leading to potential application of a selected Trichoderma strain to protect rice in a specific area of Thailand. However, I have several comments and questions here that should be taken as requested (3) and a few optional for amendment of the study:
1. Why the authors used only a single strain Z2-03 for HPLC analysis. Those others 20 Salkowski reagent negative ones, could serve (at least some of them) as negative control. To confirm the Salkowski reagent positivity or false negativity, kind of test. (requested)
2. For bio-stimulation effect on rice, similarly, more strains could be used (even as negative control). What if IAA is not only the factor responsible for competitive and stimulation activities-effects of the culture filtrates. (optional)
3. What is identity of the rest of the other Trichoderma strains? (optional)
4. In phylogenic tree there are more strains of T. breve resolved. They could be (if available) used in the study as well. (optional). What is the literature about other T. breve application as biocontrol or bio-pesticide agents? This part in Discussion is missing. (requested)
5. How much volume of the broth (culture filtrate) would be then (hypothetically) used for rood or foliar application in real conditions per hectare?
6. To use a potato dextrose broth as a solo control is questionable (as it can also possess the stimulation activities). Thus, a second crucial control = a pure water, should be used here, as well. This is rather a substantial flaw in the study to compare the effect of CF vs normal non-stimulated conditions on germination and seedlings development. I would recommend, thus, to perform this part again with a water as an additional control to have complete picture of bio-stimulation effect of CF of T.breve Z2-03.
Other comments:
Line: 220 please write Synthetic nutrient agar (SNA)
Line: 321 (Figure 4, F and G) the scale bars are unreadable, make them bigger
Line: 472 PDI rewrite as percentage of disease incidence
Well, I would strongly suggest to add (1) more strains into HPLC for presence/absence IAA confirmation - the best, all of them. (2) add water as control for bio-stimulation effect of culture filtrate. (3) add information about T. breve current usage as biocontrol agent (if available in the literature)
----end---
above
Author Response
Response to reviewer 2
A reviewer opinion on manuscript entitled “ Plant Growth Promotion and Biological Control Against Rhizoctonia solani in Thai Local Rice Variety “Chor Khing” using Trichoderma breve Z2-03 “ by Warin Intana et al.
In my opinion this is interesting case study leading to potential application of a selected Trichoderma strain to protect rice in a specific area of Thailand. However, I have several comments and questions here that should be taken as requested (3) and a few optional for amendment of the study:
Answer: Thank you very much for reviewing the manuscript and providing valuable feedback to improve it.
- Why the authors used only a single strain Z2-03 for HPLC analysis. Those others 20 Salkowski reagent negative ones, could serve (at least some of them) as negative control. To confirm the Salkowski reagent positivity or false negativity, kind of test. (requested)
Answer: In this study, we aim to screen Trichoderma isolates that produce indole compounds. We used the Salkowski reagent as the primary screening method. Isolates that tested positive with the Salkowski reagent would potentially produce indole compounds, including indole-3-acetic acid (IAA). These positive isolates were then selected for further confirmation of IAA production using a specific method, such as HPLC analysis. Based on our results, Z2-03 is the only isolate that tested positive with the Salkowski reagent. Therefore, we have selected Z2-03 for further bioassay.
- For bio-stimulation effect on rice, similarly, more strains could be used (even as negative control). What if IAA is not only the factor responsible for competitive and stimulation activities-effects of the culture filtrates. (optional)
Answer: In this study, we focused on an IAA-producing Trichoderma isolate due to its ability to increase growth and development in several plant species. Therefore, we selected the strain that tested positive with the Salkowski reagent and confirmed specific IAA production through HPLC analysis.
- What is identity of the rest of the other Trichoderma strains? (optional)
Answer: The Trichoderma fungus is a fast-growing organism that forms vegetative structures including hyphae, conidiophores, phialides, and conidia. It is not possible to distinguish between different Trichoderma species based on morphology alone. Therefore, a molecular study of multiple genes is necessary to identify them at the species level.
- In phylogenic tree there are more strains of T. breve resolved. They could be (if available) used in the study as well. (optional). What is the literature about other T. breve application as biocontrol or bio-pesticide agents? This part in Discussion is missing. (requested)
Answer: We have added more discussion about this “The Trichoderma breve type strain HMAS24844 was first isolated from a soil sample in China and identified based on morphology and molecular properties of tef1-a and rpb2 genes [58]. The application of T. breve has shown the highest chlorophyll index and increased plant height and leaf number of Theobroma cacao L. [59]. Furthermore, T. breve isolated from vermicompost effectively inhibited fungal growth of Colletotrichum siamense, which causes anthracnose on chili [60]. Our findings reveal that T. breve Z2-03 not only promotes plant growth, but also controls R. solani, the pathogen of rice sheath blight in the "Chor Khing" variety of rice. Although biological control of Trichoderma is widely used to suppress plant diseases, the use of fresh Trichoderma conidia in the field may be affected by environmental conditions such as high temperature, light, rain, and the shelf life of the formulation. Development of appropriate formulations, such as emulsion formulation, can support a long shelf life and be ready-to-use with a high concentration of conidia and a high dilution ratio with water [61]. However, we did not develop the appropriate formulation or evaluate cost-effectiveness in this study. To develop the appropriate formulation, testing its ability and evaluating cost-effectiveness is still necessary to verify in the near future.
- How much volume of the broth (culture filtrate) would be then (hypothetically) used for rood or foliar application in real conditions per hectare?
Answer: The typical amount of culture filtrate used is about 500-1,000 mL per plant seedling. If a large area of rice is being grown, for example, in a hectare, the application of cell-free culture filtrate should be equivalent to using about 1,400 liters of water for plant growth.
- To use a potato dextrose broth as a solo control is questionable (as it can also possess the stimulation activities). Thus, a second crucial control = a pure water, should be used here, as well. This is rather a substantial flaw in the study to compare the effect of CF vs normal non-stimulated conditions on germination and seedlings development. I would recommend, thus, to perform this part again with a water as an additional control to have complete picture of bio-stimulation effect of CF of T. breve Z2-03.
Answer: We have performed experiment of DW, PDB and cell free culture filtrate of T. breve Z2-03 as showed in Fig. 2.
Other comments:
Line: 220 please write Synthetic nutrient agar (SNA)
Answer: We have revised as “synthetic nutrient agar (SNA)”.
Line: 321 (Figure 4, F and G) the scale bars are unreadable, make them bigger
Answer: We have revised scale bar of Fig. 4 F and G.
Line: 472 PDI rewrite as percentage of disease incidence
Answer: We have revised as PDI…
Well, I would strongly suggest to add (1) more strains into HPLC for presence/absence IAA confirmation - the best, all of them. (2) add water as control for bio-stimulation effect of culture filtrate. (3) add information about T. breve current usage as biocontrol agent (if available in the literature
Answer: Thank you for providing valuable comments and suggestions to improve this manuscript.
(1) As mentioned earlier, we use the Salkowski reagent for primary screening of indole compound-producing Trichoderma. Only the Z2-03 isolate tested positive with the Salkowski reagent, and Z2-3 was then subjected to quantify IAA using HPLC.
(2) We included distilled water as the control group in the experiment, as demonstrated in Fig. 2.
(3) The discussion about T. breve has been added to the discussion section.
Round 2
Reviewer 2 Report
my major comments e.g. to use water as a control and T.breve from literature were added. teh authors made additional efficiency study with water control. well, I believe, the mansucript is now suitable for publication. it is very good study.
LIne 25 it is " The cell-free CF " rewrite as "The cell- free culture filtrate of potato dextrose broth (CF)
Line 26 it is (DW....)" use (distilled water ...)
Line 127 it is ".. was cultured in PDB" used was cultivated in potato dextrose broth (PDB)
Author Response
Major comments
my major comments e.g. to use water as a control and T.breve from literature were added. teh authors made additional efficiency study with water control. well, I believe, the mansucript is now suitable for publication. it is very good study.
Answer: Thank you for reviewing the manuscript and providing valuable comments to enhance its quality.
Detail comments
LIne 25 it is " The cell-free CF " rewrite as "The cell- free culture filtrate of potato dextrose broth (CF)
Answer: We have revised as suggested.
Line 26 it is (DW....)" use (distilled water ...)
Answer: We have revised it as “distilled water…
Line 127 it is ".. was cultured in PDB" used was cultivated in potato dextrose broth (PDB)
Answer: We have revised it as “…was cultivated in potato dextrose broth (PDB).